# Statistical Bioinformatics to Uncover the Underlying Biological Mechanisms That Linked Smoking with Type 2 Diabetes Patients Using Transcritpomic and GWAS Analysis

**DOI:** 10.3390/molecules27144390

**Published:** 2022-07-08

**Authors:** Abu Sayeed Md. Ripon Rouf, Md. Al Amin, Md. Khairul Islam, Farzana Haque, Kazi Rejvee Ahmed, Md. Ataur Rahman, Md. Zahidul Islam, Bonglee Kim

**Affiliations:** 1Department of Statistics, Jagannath University, Dhaka 1100, Bangladesh; ripon1969@gmail.com; 2Department of Computer Science & Engineering, Prime University, Dhaka 1216, Bangladesh; ahmedalamin2357@gmail.com; 3Department of Information & Communication Technology, Islamic University, Kushtia 7003, Bangladesh; mdkito51@gmail.com; 4Department of Biotechnology and Genetic Engineering, Faculty of Biological Sciences, Islamic University, Kushtia 7003, Bangladesh; linktofarzana@gmail.com; 5Department of Pathology, College of Korean Medicine, Kyung Hee University, Hoegidong Dongdaemungu, Seoul 02447, Korea; kazirejveeahmed@gmail.com; 6Korean Medicine-Based Drug Repositioning Cancer Research Center, College of Korean Medicine, Kyung Hee University, Seoul 02447, Korea

**Keywords:** Type-2 diabetes, smoking, bioinformatics, association, GSEA, pathway, orthology

## Abstract

Type 2 diabetes (T2D) is a chronic metabolic disease defined by insulin insensitivity corresponding to impaired insulin sensitivity, decreased insulin production, and eventually failure of beta cells in the pancreas. There is a 30–40 percent higher risk of developing T2D in active smokers. Moreover, T2D patients with active smoking may gradually develop many complications. However, there is still no significant research conducted to solve the issue. Hence, we have proposed a highthroughput network-based quantitative pipeline employing statistical methods. Transcriptomic and GWAS data were analysed and obtained from type 2 diabetes patients and active smokers. Differentially Expressed Genes (DEGs) resulted by comparing T2D patients’ and smokers’ tissue samples to those of healthy controls of gene expression transcriptomic datasets. We have found 55 dysregulated genes shared in people with type 2 diabetes and those who smoked, 27 of which were upregulated and 28 of which were downregulated. These identified DEGs were functionally annotated to reveal the involvement of cell-associated molecular pathways and GO terms. Moreover, protein–protein interaction analysis was conducted to discover hub proteins in the pathways. We have also identified transcriptional and post-transcriptional regulators associated with T2D and smoking. Moreover, we have analysed GWAS data and found 57 common biomarker genes between T2D and smokers. Then, Transcriptomic and GWAS analyses are compared for more robust outcomes and identified 1 significant common gene, 19 shared significant pathways and 12 shared significant GOs. Finally, we have discovered protein–drug interactions for our identified biomarkers.

## 1. Introduction

Type 2 diabetes(T2D) is a chronic metabolic disease defined by insulin insensitivity corresponding to impaired insulin sensitivity, decreased insulin production, and eventually, failed beta cells in the pancreas [1,2]. So, T2D occurs when the body cannot effectively utilise insulin [3]. According to current forecasts, the total number of diabetes patients would be greater than 50% from 2017 to 2045, resulting in over 693 million or 0.693 billion people with diabetes. Therefore, the health expenditure will be about 850 billion US dollar [4]. It is unpleasantly surprising that in diabetes patients, more than 95%, have T2D [5]. In consequence, diabetes was responsible for about five million deaths in 2017 among people aged 20 to 99 years over the world [4].

It is clinically statistically shown that professional smokers are more likely to develop T2D with a 30–40% higher chance than nonsmokers [6]. Another research study demonstrated the smoking prevalence in China by examining 1658 people, including 621 (37.5%) non-smokers and 1037 (62.5%) people actively smoking. Whereas diabetes was found to be more related to active smoker than the non-smoker [7]. Not only that, both men and women smokers were much more likely to establish T2D than those who never smoked [8,9]. Moreover, diabetes patients with a smoking history have increased risk of developing complications such as heart and kidney disease, and nerve damage, as well as alleviated blood supply to the legs and feet [6]. The reason for the relation between smoking and T2D is the dysfunction of beta-cells, where smoking may harm the act of beta-cells [10,11]. Similarly, it is also visible that the beta-cells are harmed in T2D patients [2]. Due to the damage of beta-cells, there is an association between the increase of T2D and insulin resistance, which can be altered by smoking in both a direct and an indirect way [12,13]. According to a new theory, Cigarette smoking was associated with lower vitamin D levels [14,15], where 83.2% of patients with T2D have vitamin D deficiency, which is a significant number [16]. Another represented that a higher casualty of T2D is connected with lower HDL cholesterol levels [17,18], and smoking cigarettes is also linked to lower HDL cholesterol levels [19,20]. In addition, increased (triglycerides) TG levels were found to be substantially linked to the development of T2D [21,22], whether smoking helps increases triglycerides [23].

The development of T2D can be amplified by regular smoking, as well as many complications which can be developed in T2D patients with a smoking history. Hence, we have tried to retrieve common molecular mechanisms that exist between T2D and smokers. Molecular mechanisms and relationships underlying the T2D-smoker are yet unclear in medical endocrinology [24], but they are still of significant interest. Moreover, there is still a lack of bioinformatics studies regarding the relationship between T2D and smoking. The goal of our study is to warn the active smokers that they can develop T2D and to help the drug developers so they will be able to eliminate the risk of many complications in T2D patients due to smoking. Therefore, this inspired us to apply a bioinformatics-based systemic pipeline to analyse the gene expression data of T2D patients and smokers to obtain an insightful understanding of their relationship. To comprehend molecular causes of biological disease and condition, differentially expressed genes (DEGs) [25] should be identified for further system biology analysis. In this study, RNA-seq datasets [26] (transcriptomic data [27]) were utilised to discover DEGs shared by T2D and smoker. We have used the DESeq2 package [28] which is frequently used for differential gene expression analysis of RNA-seq count data. We identified commonly dysregulated genes, pathways and Gene Ontology (GO) protein–protein interaction (PPI) [28], hub-proteins [29], Transcription Factor Gene Interactions [30], Gene miRNA Interaction [31], Protein–Drug Interaction [32]; that are associated with T2D and smoker using a systems biology approach. Finally, we have analysed common genes, pathways and GO using GWAS datasets [33]; later, we discuss and compare with transcriptomic analysis. We have represented the methodology of our hypothesis as shown Figure 1.

## 2. Materials and Methods

### 2.1. Datasets Employed in This Study

We used data from the Gene Expression Omnibus (GEO) of the National Center for Biotechnology Information (NCBI) [34]. Datasets for our work are also available in GREIN, a web-based interactive platform that offers simple substitutes for exploring and analysing GEO RNA-seq data [26]. GREIN was utilized in many more studies to perform differential expression analysis of their RNA-seq dataset [35]. Many datasets were returned for each disease while searching in NCBI; however, most of them were rejected since they did not fulfil the sample size where the size should have been 6; similarly, neither replicate datasets or RNAseq datasets which were not taken from homo sapien organisms, as well as datasets, do not include both a Case and Control. In six samples, there should be at least three case and three control. For this type of study, we looked for datasets with the lowest amount of bias and noise. Here, after many searches, we have selected datasets that overcome any noise as much as possible for this analysis. This procedure selected two datasets that are suitable for our investigation and are very relevant to T2D and smoking.

Accession numbers of the datasets, which is human gene expression, are GSE106177 [36] and GSE47718 [37]. T2D (GSE106177) dataset is collected from the DNA of human primary cardiac mesenchymal cells (CMSC) from seven diabetic (D) donors and seven non-diabetic (ND) donors were analyzed. The genomic DNA of human CMSCs isolated from diabetes donors had an accumulation of 5-methylcytosine, 5-hydroxymethylcytosine, and 5-formylcytosine, as determined by quantitative global analysis, methylated and hydroxymethylated DNA sequencing, and gene-specific GC methylation detection. The smoking datasets (GSE47718) are generated from the human airway basal cell (BC) transcriptome of seven smokers and seven non-smokers. On 19q13.2 (CYP2F1 and RASGRP4), two airway epithelium genes were also identified to study hypermethylation in smokers in comparison with non regular smokers (shown in Table 1). As well as this, BC of the case and control were compared utilizing massive parallel RNA sequencing.

### 2.2. Preprocessing of Raw Counts and Its Differentially Expression Analysis

The GREIN Gene Expression Omnibus(GEO) provided us with gene expression RNA-sequence datasets. The DESeq2, an R package, was used to identify differentially expressed genes (DEGs) of T2D and Smoking related RNA-Seq count data [28]. In DESeq2, the geometric mean of every single gene was calculated among all samples to normalize the data. Then, DESeq2 used the cook’s distance to automatically filter out lowly expressed and outlying genes. Based on two conditions such as ∥log2FC∥≥ 1 and adjusted *p*-value ≤ 0.05, the significant DEGs were filtered. By utilising the match function, we were able to find shared DEGs between T2D and smoking and create a Venn diagram (Shown in Figure 2).

### 2.3. Identification of Molecular Pathway and Gene Ontology

Many bioinformatic techniques often focus on measuring the relevance of gene set similarity with previously annotated gene sets in order to find functional pathways connected to a particular disease [38]. We used gene set enrichment analysis with EnrichR to find out its associated pathways and Gene Ontology (GO) terminologies by using the overlaying DEGs between T2D and smokers. The findings identified pathways and GOs, which would definitely help us to gain an appropriate understanding of the biological mechanism related to both conditions [39]. For biological domains, the underlying purpose of the GO project is to develop a structured gene dictionary that may be used to explain gene-products in any living life form. In terms of GO terminology, there are three categories: biological process (BP), cellular component (CC), and molecular function (MF) [40]; however, we only considered the BP for our study. Pathways demonstrate a crucial role in how organisms respond to stimuli. In life science, pathway analysis is commonly used to help researchers understanding the high-throughput biological data through their underlying molecular mechanisms. It may also describe the relation of diseases or conditions to each other [41].

### 2.4. GWAS Data of Type 2 Diabetes Mined to Compare with the GWAS Data of Smoker

Researchers use genome-wide association studies (GWAS) to examine a group of genetic mutants which are linked with a certain illness or condition. During this procedure, the genomic sequences of a large number of patients were collected and evaluated so that we could determine the DNA polymorphisms that are present in the genomes of the patients. We have collected GWAS data from both T2D patients and smokers. We have collected the GWAS data from two well-known databases such as PheGenI [42] and GWAS Catalog [43]. Both the databases created using previous studies that incorporated GWAS results published earlier. We have only considered data for T2D patients and smokers. Moreover, we have filtered the most significant data by a threshold *p*-value that is less than 1.0 × 10^−8^. We also identified the GOs and pathways associated with the identified common significant genes from GWAS analysis. Finally, we have compared the transcriptomic analysis result with the GWAS analysis to better understand and relate the common biological outcomes by both studies.

## 3. Results

### 3.1. Gene Expression Analysis of Transcriptomic Data

We have collected RNA-Seq data from Grein or NCBI to examine the impact of gene expression on T2D patients and Smokers. As shown in Table 1, “Illumina Next Seq 500 (Homo sapiens)” GEO platform has provided the T2D data, and the “Illumina HiSeq 2000 (Homo sapiens)” has provided the smoking data. The T2D RNA-seq data have been collected from “Human cardiac mesenchymal cells”, and the smoking RNA-seq data have been collected from “Airway Basal Cells”. The GEO accession ID-GSE106177 is selected for T2D [36], and GEO accession ID-GSE47718 is selected for Smoking [37]. In T2D, 14 samples were included, seven for the case and seven for control. The total differentially expressed (DE) genes for T2D is 18,619, which is found after differential expression analysis; which is also called generated signature data. Then, two conditions considered such as “Adjusted PValue” and “Log2 Fold-Change” on signature data. We have found 1367 significant genes after applying the condition whereas Adjusted PValue is less than 0.05 and abs(LogFC) is greater than or equal 1.0. Among significant genes, 768 is Up-regulated genes, and 599 is downregulated genes. On the other hand, a total of 17 samples are included in the Smoking dataset, whereas 7 for the case(regular-smokers) and 10 for control(non-smokers). After generating signature data, we have found 21178 DE genes. Then applied the same conditions that applied to the T2D, and we have found 962 significant genes. Among these, 682 is positively expressed (Up-regulated), and 280 is negatively expressed (downregulated).

Then, we have compared the upregulated genes of T2D with the upregulated genes of smokers. Similarly, we have compared the downregulated genes of both conditions. Between T2D and Smoking, the number of upregulated common genes are 27 and the number of downregulated common genes are 28 (Shown in Figure 3). The most significant shared up regulated genes are *TMEM178B*, *TAPBP*, *ZNF469*, *SPP1*, *ARHGAP27*, *XIST*, *HCN2*, *TDRD12*, *METRN*, *MEGF6*, *KIT*, *IDUA*, *DUSP8*, *ABR*, *C11orf96*, *ZBTB22*, *GLIS2*, *EPPK1*, *CPXM2*, *PRRC2A*, *NTRK1*, *HLA-C*, *LOC101928994*, *VGF*, *ICAM5*, *FSCN2* and *KCNJ12*. Moreover, the most significant shared down regulated genes between T2D and Smoking: *LRRCC1*, *EPHA7*, *ARHGAP20*, *SELL*, *FLOT1*, *PCDH18*, *CFI*, *ARHGAP11A*, *CENPH*, *KRBOX1*, *MYB*, *PRIM1*, *MICA*, *HMSD*, *CDCA7*, *KIFC1*, *HLA-DPA1*, *ZNF385D*, *GPR158*, *SMN1*, *HLA-DPB1*, *MGP*, *CERS6-AS1*, *PSMB8*, *PCYT1B*, *E2F2*, *TCF19* and *RNVU1-4.*

### 3.2. Pathway and GO Related Functional Association Analysis

In pathway analysis, we have used five different databases named BioCarta, Elsevier Pathway, KEEG, Reactome and WikiPathway. Then, we have looked for highly expressed pathways by utilizing shared DEGs between T2D and smoking. These pathways were then divided into functional groups for analysis. Initially, we got 573 signalling pathways linked with both disease and condition. Manual curation was then utilised to abate the number of pathways. Pathways with *p*-values below 0.05 are considered. Therefore, we got 169 most significant signalling pathways. Finally, we sorted the pathways in ascending order depending on *p*-value. The top 20 pathways were represented in Figure 4 that were linked to T2D and smoking condition.

The GO approach uses the biological process (BP), cellular component (CC), and molecular function (MF) to categorise them into functional categories, but we only consider the BP database of GO terminologies. At first, 489 GO terms found common between T2d patients and smokers. After that, the terms with a *p*-value less than 0.05 were included as the most significant GO terms. A total of 137 GO terminologies are found as the mostly enriched GO terms between the conditions. Figure 5 summarises the top 20 most significant GO terms of BP between T2D and smoking.

### 3.3. Protein–Protein Interactions (PPIs) Analysis

STRING [44], a database, was handled to perform protein–protein interactions (PPI). In Cytoscape [45], the PPI network has been processed and analysed. Based on proteins encoded by the common DEGs between T2D and Smoking, we developed a PPI network on the STRING. In Cytoscape, this is prepared and assessed, and the results are shown in Figure 6. Protein subnetworks that are shared by two or more diseases are known to be linked to each other. The highly interacting proteins were discovered by PPI analysis using topological characteristics, such as degree larger than 15°. The CytoHubba [46], a plugin in Cytoscape, was installed and operate to find the highly linked hub proteins in the PPI network using degree and Maximal CliqueCentrality (MCC) and BottleNeck algorithm. These newly discovered hub proteins may be beneficial as therapeutic targets, but further study is needed to understand their functions. As mentioned earlier, hub proteins are identified using two algorithm named MCC and BottleNeck, and represented in Figure 7A,B. By using the MCC algorithm we have found a total of 23 hub proteins where the top 10 genes are highly significant marked as red, orange and yellow colors. In addition, by using the BottleNeck algorithm, we have found a total of 20 hub proteins where 10 are shown as the most significant (marked as yellow, red and orange colors). In both algorithms, we have found a total of 13 most significant hub proteins that are unique and they are: KIFC1, TAPBP, HLA-DPA1, HLA-DPB1, PSMB8, SPP1, KIT, MYB, MICA, PRIM1, TCF19, HLA-C and ZBTB22. Among them seven proteins are shared by both algorithms that are: KIFC1, TAPBP, HLA-C, HLA-DPB1, HLA-DPA1, KIT and PRIM1 as well as most significant.

### 3.4. Identification of Transcriptional and Post-Transcriptional Regulators of the Differentially Expressed Genes

For TFs, protein–drug interactions, and gene–microRNA interactions, we have used Net-workAnalyst tools [47]. Transcription factors (TFs) are proteins that control gene expression in all living organisms. In every cell process, TFs perform a critical function [48]. Gene expression is regulated in the post-transcription stage by non-coding short RNA molecules called miRNAs. Protein–drug interaction studies are critical to understanding the structural characteristics required for ligand affinity [49,50]. The TFs-Gene network was constructed using the ChIP-X [51] and JASPAR database [52]. The TarBase [53] and miRTarBase database [54] were utilized to develop Gene–miRNAs interaction network using NetworkAnalyst. Moreover, protein–drug interactions are constructed using the DrugBank database [55]. Figure 8 visually represents the TF–Gene Interactions, and the regulator genes are: *ARHGAP27*, *E2F2*, *KIT*, *EPHA7*, *TAPBP*, *KIFC1*, *ARHGAP11A*, *SPI1*, *POU5F1*, *NANOG*, *SOX2*, *METRN*, *TCF19*, *SMN1*, *TDRD12*, *MGP*, *E2F1*, *GATA2*, *NFKB1*, *NFIC*, *SRF*, *YY1*, *TFAP2A*, *FOXL1*, *TP53*, *POU2F2*, *FOXC1*, *PRRC2A*, *VGF*, *CDCA7*, *SPP1*, *FLOT1*, *ABR* and *MYB*. Figure 9 visually represents miRNA’s genes interactions that are: mir-146a- 5p, mir-34a-5p, mir-27a-3p, mir-129-2-3p, mir-124-3p, mir-193b-3p, let-7b- 5p, mir-26b-5p, mir-31-5p, mir-93-5p, mir-331-3p. Finally, Figure 10 visually represents the Protein–Drug Interactions for smoker and T2D complications and common genes are: KIT and NTRK1. The genes are related to drugs called Ponatinib, Dasatinib, Pazopanib, Sorafenib, ABT-869, OSI-930, MP470, XL184, XL820, Nilotinib, Phosphonotyrosine, Lenvatinib, Sunitinib, Imatinib, Regorafenib and Amitriptyline.

### 3.5. GWAS Analysis of Type 2 Diabetes with Smoker and Comparison with Transcriptomic Analysis

GWAS analysis was performed to find the most significant genes for both T2D and smokers. We employed a threshold *p*-value which is less than 1.0 × 10^−8^. After that, we found 595 significant genes for T2D and 535 significant genes for smokers. Then, we compared both significant genes to obtain the candidate genes associated with both conditions. There are 57 candidate genes (Shown in Figure 11) that are shared by both conditions: *GNPDA2*, *ITPR2*, *JAZF1*, *MSRA*, *ZC3H4*, *NRXN3*, *MAP2K5*, *TCF7L2*, *HSD17B12*, *CCND2*, *AUTS2*, *FTO*, *MC4R*, *ANKRD55*, *PPARG*, *TMEM18*, *VEGFA*, *TFAP2B*, *CTTNBP2*, *PINX1*, *C5orf67*, *LYPLAL1*, *MIR4432HG*, *COBLL1*, *ADAMTS9-AS2*, *NFAT5*, *CMIP*, *ALDH2*, *ZFHX3*, *SEC16B*, *TSEN15*, *SIX3*, *BCL11A*, *ZBTB20*, *FGFR4*, *HMGA1*, *CALCR*, *BDNF*, *FAIM2*, *TRIB1*, *ARID5B*, *ADAMTS9*, *LINGO2*, *EHMT2*, *ZBTB38*, *THAP12P9*, *Metazoa_SRP*, *CCND2-AS1*, *HLA-C*, *SPPL3*, *LINC02537*, *RFLNA*, *TSEN2*, *ZC3H11B*, *RSPO3*, *NYAP2* and *CUX2*. Moreover, we have used the identified biomarker genes from GWAS analysis in order to find the signalling pathways and GO terms. We have used the same tools as transcription profiles for these analysis. In GWAS analysis, we have got 839 signalling pathways and 802 GOs shared by both T2D and smoker. Moreover, Figure 12 and Figure 13 represent the top 20 GOs and pathways respectively among them. We have also used the same condition (*p*-value ≤ 0.05) to distinguish the most significant pathways and GOs. Thus, the number of pathways reduced to 196; similarly, GOs reduced to 300. A comparison performed between transcriptomic and GWAS analysis where we have found only one biomarker gene (HLA-C) common between two studies. We have also compared significant pathways from both transcriptomic and GWAS analysis and found 19 significant pathways are shared by both studies: Genes with Mutations Associated with Psoriasis, Proteins Involved in Ulcerative Colitis, Proteins Involved in Psoriatic Arthritis, Proteins Involved in Neuroblastoma, Proteins Involved in Spontaneous Abortion, Proteins with Altered Expression in Osteoarthritis, NTRK FOXO/MYCN Signalling, NTRK1/2/3 Acetylcholine Production, Receptors and Adaptor Proteins Activated in Cancer, Proteins Involved in Glioma, Kaposi sarcoma-associated herpesvirus infection, MAPK signalling pathway, PI3K-Akt signalling pathway, Endosomal/Vacuolar pathway, VEGFR2 mediated cell proliferation, PI3K-Akt signalling pathway WP4172, Allograft Rejection WP2328, Pathways Regulating Hippo Signalling WP4540 and NOTCH1 regulation of endothelial cell calcification WP3413. Similarly, 12 significant GOs are shared by both studies: regulation of neuron apoptotic process (GO:0043523), positive regulation of phosphorylation (GO:0042327), antigen processing and presentation of endogenous peptide antigen via MHC class I via ER pathway (GO:0002484), nerve growth factor signalling pathway (GO:0038180), negative regulation of sprouting angiogenesis (GO:1903671), positive regulation of bone resorption (GO:0045780), positive regulation of DNA-binding transcription factor activity (GO:0051091), ovarian follicle development (GO:0001541), neurotrophin TRK receptor signalling pathway (GO:0048011), positive regulation of histone H3-K4 methylation (GO:0051571), positive regulation of cell projection organization (GO:0031346) and sympathetic nervous system development (GO:0048485).

### 3.6. Validating Potential Biomarker Targets Using Earlier Literatures

We investigated the previously published literature to verify the presence of biomarker genes that are found both in smokers and T2D (Shown in Table 2). The expression of SPP1 can be boosted by smoking cigarettes [56] and SPP1 expression similarly raised in patients with T2D [57]. Smoking has been linked to MICA [58]. PSMB8 was found linked with regular smokers [59]. The protein PSMB8 was observed in the proteosome that is responsible for ATP-dependent protein degradation, which is a direct reason for developing T2D [60]. Patients with T2D had a higher level of XIST expression than those without diabetes [61]. Type 2 diabetes risk is substantially increased by DUSP8 expression in individuals [62]. Patients with T2D are more likely to experience ABR abnormalities than those without the disease [63]. In both type 2 diabetes and atherosclerosis, the protein PRRC2A was found to be present [64]. Compared to non-smokers, heavy smokers show a substantial decrease in the transcription of PRRC2A genes [65]. The NTRK1 has been identified in T2D [66]. It is possible that the EPHA7 gene is linked to smoking [67]. T2DM may be impacted by SELL [68]. T2D patients are linked with PRIM1 deficiency [69]. Carcinogenic pathways caused by PRIM1 may be influenced by factors in the tumor microenvironment, such as smoking (nicotine) [70]. There are genes called KIFC1 that are connected with a person’s smoking status [71]. ZNF385D exhibited increased levels of methylation in people who are currently smoking [72]. In T2DM, the MHC-class II molecules, particularly HLA-DPB1, which encode human leukocyte antigens, were dramatically increased [73].

## 4. Discussion

There is no doubt that T2D is a major threat to humanity. People who smoke cigarettes have a 30–40% increased chance of acquiring T2D in comparison with nonsmokers [6]. Moreover, we have highlighted through a study that many complications can be developed such as kidney and heart disease in T2D with smoking addiction [6]. So, we can say that it is possible that active smokers may easily develop T2D and it is also possible that T2D patients with a smoking history may develop many complications or diseases. Therefore, our study aims to extract genetic correlations between smoking and T2D. Furthermore, our predicted drugs might be the potential therapeutics for T2D patients with a smoking history. Not only that, but also physicians would inspire the patients to stay away from smoking. This would definitely eliminate the risk of developing T2D in individuals or other complications caused in T2D patients. The bioinformatics approach provides a comprehensive understanding of the molecular mechanisms in disease progression. In this study, the investigation was carried out on both transcriptomic and GWAS profiles of smokers and T2D.

To investigate if there was any significant dysregulation, we have performed Differential Expression Analysis (DEA) followed by identification of common genes including up or downregulated genes (shown in Figure 2 and Figure 3, respectively), gene ontology (GO) (Shown in Figure 5), pathways (represented in Figure 4), protein–protein interactions (shown in Figure 6), hub–protein interactions (shown in Figure 7), transcription factor gene interactions (represented in Figure 8), gene–miRNA interactions (shown in Figure 9) and protein–drug interactions (represented in Figure 10) using transcriptomic profiles of smoking and T2D. As well as, we have identified common genes, GOs and pathways using GWAS profiles of smoking and T2D (respectively, shown in Figure 11, Figure 12 and Figure 13). GWAS data analysis provides a more robust understanding of our hypothesis as well as compared with transcriptomic profiles. The T2D transcriptomic dataset was collected from T2D patients and non-diabetic healthy individuals, and the smoking dataset was collected from smokers and nonsmokers (as shown in Table 1). We also verified our results by previous literature published in various journals. The flow diagram of our methodology has been visually represented and outlined with proper direction in Figure 1.

First of all, we have focused on four pathways, namely PI3K/AKT signalling pathway, MAPK signalling pathway, Endosomal/Vacuolar pathways and Allograft Rejection pathway, which are found by the shared profile of T2D and smoker. These pathways were resulted from both transcriptomic and GWAS analysis.

T2D development is significantly influenced by the MAPK signalling pathway [74], where cigarette smoke particles may activate MAPK pathways [75]. The Allograft Rejection pathway is also activated in T2D patients [76], where Allograft Rejection is more likely to active in people with a history of smoking cigarettes [77,78]. PI3K/AKT pathway damaged in various tissues of the body leads to the development of T2D as the result of insulin resistance [79], and tobacco smoke exposure can activate PI3K/Akt pathway [80]. Endosomal/Vacuolar pathways is presented in T2D [81]. Hence, in vitro, researchers can work on this pathway to make resistance to T2D. Therefore, the continuous inactive stage of the Allograft Rejection pathways, PI3K/Akt pathway and MAPK pathway may be possible by quitting smoking at an early period or taking drugs to alter the current active status of the pathways in T2D patients. Ultimately, avoiding smoking is highly recommended for both control (absence of T2D) and T2D patients so that the pathways do not influence by smoking. As far as we know, no bioinformatics approach had previously described these pathways and to the best of my knowledge, other pathways identified in our study are not specified by any previous literature for developing T2D. Yet those pathways may be potential drug targets that paved the way for further research and analysis.

For significant common genes, we have focused on certain genes named SPP1, PSMB8, PRRC2A, and PRIM1 that are found from transcriptomic analysis. These genes are highly connected with each other, which also resulted in hub–protein analysis (as shown in Figure 7). These genes have been picked up based on the previously published literature related to both smokers and T2D patients (Shown in Table 2). SPP1 has been linked to an increased risk of long-term effects of diabetes and has been found to be elevated in T2D patients [57]. Where, SPP1’s expression in induced sputum can be raised by smoking cigarettes, and the amount of SPP1 is also increased in induced sputum [56]. PSMB8 was found to be associated with insulin-dependent Diabetic Mellitus [82]. Moreover, it is highly expressed among smokers [59]. The PRRC2A gene was responsible for the development of insulin-dependent diabetes mellitus [83]. It was found that nicotine therapy increased the expression of PRIM1 protein [70]. Moreover, PRIM1 mutation has also been recorded for developing T2D [69]. As the specified genes such as PRIM1, SPP1, PSMB8 and PRRC2A are activated in T2D patients, it is essential to remove the mutation of these genes to rehabilitate. However, it is smoking that regulates the activation of these genes in individuals as well. Thus, the effect of smoking is a serious issue for one’s health and needs to avoid in the first place. As far as we know, these genes were not analysed previously in any bioinformatics study.

In protein–drug interactions, we have discovered two genes named KIT and NTRK1 that are connected to 16 drugs for T2D with a smoking history (shown in Figure 10). KDM6A binding to the NTRK1 promoter because of YY1 and subsequent TRKA overexpression led to imatinib resistance [84]. Moreover, it has been demonstrated that imatinib improves glycemic control in diabetics [85]. According to this information, imatinib is connected to the NTRK1 gene, and it helps to control T2D. Comorbid diabetes mellitus, IFG, or IGT can be treated with sorafenib, a safe and effective therapeutic option [86]. In addition, sorafenib inhibits the Imatinib-Resistant KIT. As well, stable Ba/F3 KIT mutants recapitulated the genotype of imatinib-resistant individuals with primary and secondary KIT mutations [87]. So, this sorafenib drug can resist T2D. It was found that sunitinib lowered insulin requirements [88]. In addition, tyrosine kinase inhibitors (TKIs), such as Sunitinib and other TKIs, have been demonstrated to lower blood glucose levels in T2D patients [89]. Sunitinib was administered to patients with KIT overexpression, and responses were evaluated using the Response Evaluation Criteria in Solid Tumors (RECIST) and those with KIT mutations are more responsive to Sunitinib. As well, sunitinib may be superior to other KIT inhibitors due to its ability to inhibit VEGFR1, VEGFR2, VEGFR3, and platelet-derived growth factor receptor (PDGFR) [90]. So, inhibiting mutant-like KIT in metastatic melanoma with nilotinib and dasatinib is becoming increasingly popular. Apart from binding and inhibiting ABL’s kinase domain, nilotinib affects KIT and PDGF receptor kinases more effectively than imatinib [91]. In an accelerated-phase, CML patient with T2D, dasatinib reduced hyperglycemia rapidly [92]. The resulted drug is found from a shared analysis of T2D and smoking. Although the drug is only validated for T2D as per previous literature, it may also mediate the negative consequence that occurred from smoking. Therefore, our predicted drug may be well performed for T2D with a smoking history.

We have tried to validate all our findings with previous literature. As it was not possible, therefore, there is still scope for further in vivo and in vitro level research. Based on the findings, the doctor/physician may recommend that active smokers abstain from the bad habit to minimise the likelihood of T2D. Furthermore, smoking-controlled complications in T2D patients are a major concern for T2D treatment. Thus, avoiding smoking is highly recommended for T2D patients. Moreover, the pharmaceutical sectors may develop drugs depending on the resulted chemical compound for the treatment of T2D, which may also reduce the evolved smoking effect. For the limitations of this study, it should be noted that our selected datasets have fewer samples. Age, sex, ethnicity and other relevant attributes were not considered for this study. Hence, further validation is required to thoroughly analyse the biological relevance related to the stated problem in this study.

## 5. Conclusions

To obtain better knowledge about the risk of smoking, we have conducted a bioinformatics analysis in this study. We have found that smokers have a greater risk of developing Type 2 diabetes, and T2D patients could quickly evolve many complications due to smoking. Therefore, we have conducted various statistical analyses on both transcriptomic and GWAS profiles of active smokers and T2D patients. Our study resulted in 55 shared biomarker DEGs between smokers and T2D from the transcriptomic analysis. Whereas seven genes ( SPP1, PSMB8, PRRC2A, PRIM1, KIT, NTRK1 and HLA-C) are validated using previous studies. They are also repeated in multiple analyses such as GWAS analysis, protein–protein interaction, protein–drug interactions, and transcriptional and post-transcriptional Regulators. Signalling pathways such as PI3K/AKT signalling pathway, AMPK signalling pathway, Endosomal/Vacuolar pathways and Allograft Rejection are also verified using published literature. However, still, there are many novel findings that make way for further analysis by the researchers. Finally, our study suggests that people should quit smoking as soon as possible in order to reduce the risk of being diagnosed with type 2 diabetes. Moreover, type 2 diabetes patients must quit smoking to avoid other complications/diseases and take medication accordingly.

## Figures and Tables

**Figure 1 molecules-27-04390-f001:**
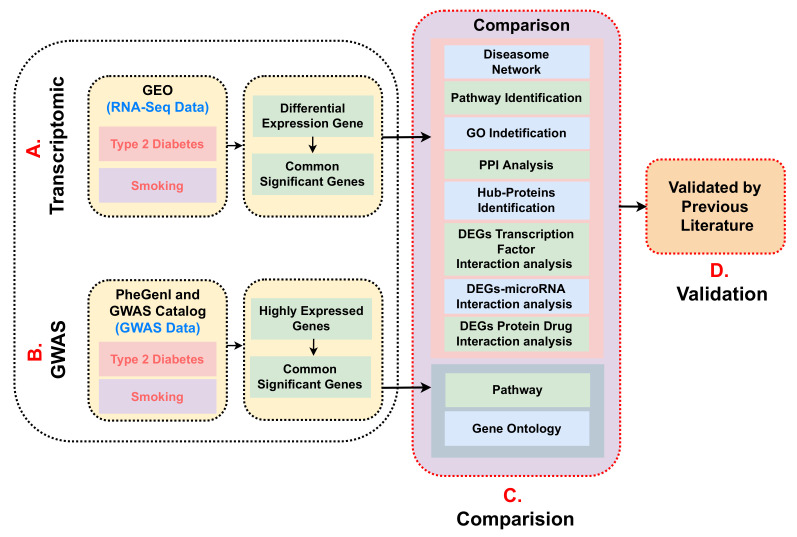
The figure demonstrate the working flow of our proposed methodology. **A**. Collected Transcriptomic datasets from GEO for Type 2 diabetes and Smoking. Generate DEGs and find out shared significant genes between both conditions **B**. Collected GWAS datasets from PheGenl and GWAS Catalog databases for Type 2 diabetes and Smoking. Generate DEGs and find out common significant genes between the conditions. **C**. Statistical analyses on Transcriptomic data to identify Pathway, Gene Ontology, Diseasome Network, PPI Analysis, Hub-Proteins, DEGs Transcription Factor Interaction, DEFs–microRNA Interaction, and Protein–Drug Interaction. Comparison between Transcriptomic and GWAS analysis based on Pathway and Gene Ontology. **D**. Validated the results by previous literature to find the biological relevance.

**Figure 2 molecules-27-04390-f002:**
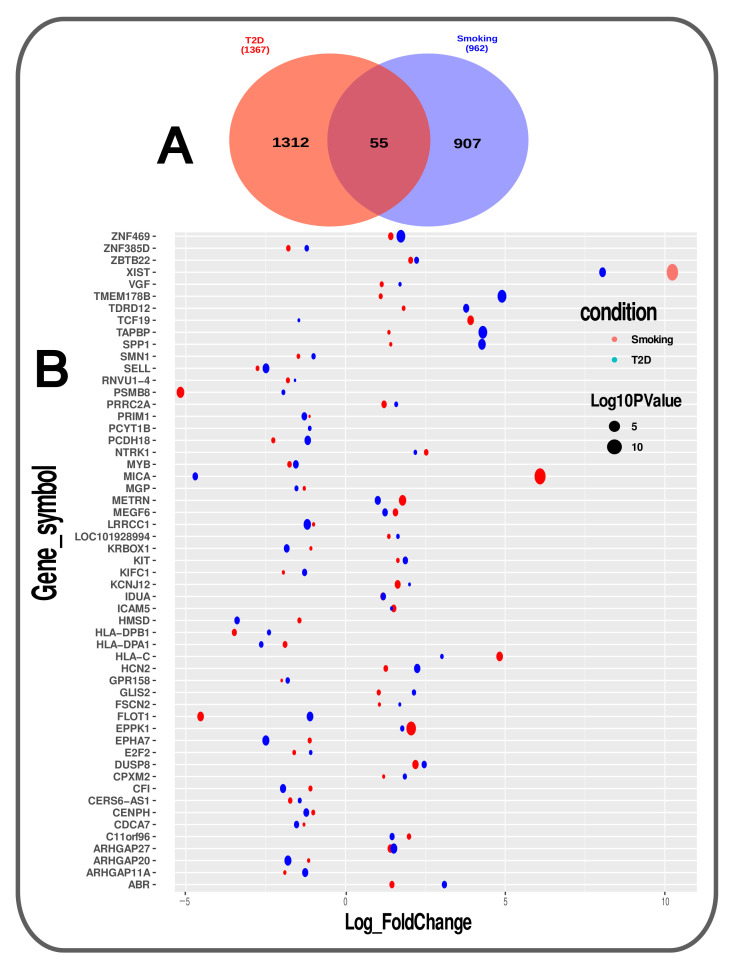
The figure represented the summary of transcriptomic analysis. **A**. Common biomarker Genes between T2D and Smoking using a venn diagram **B**. Bubble plot of common biomarker genes and their associated adjusted *p*-value and Log2 fold change.

**Figure 3 molecules-27-04390-f003:**
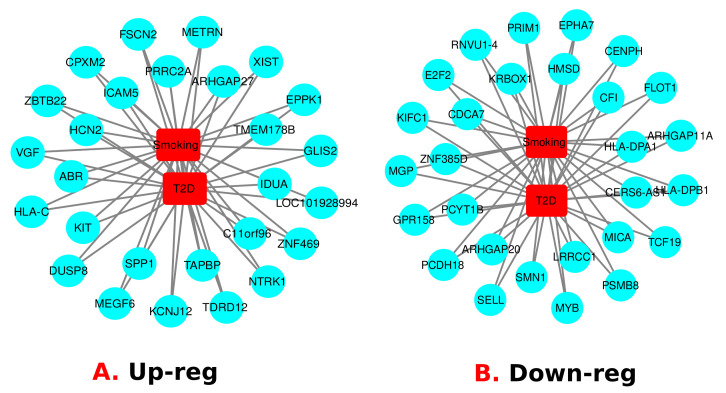
Upregulated and downregulated genes between T2D patients and smokers are represented separately.

**Figure 4 molecules-27-04390-f004:**
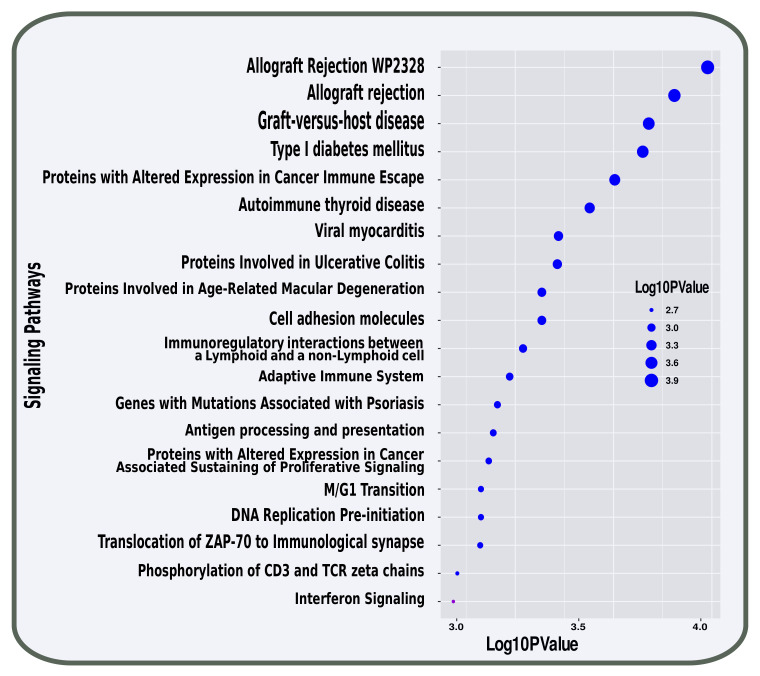
Top 20 pathways are represented using a bubble plot that are associated with both conditions in transcriptomic analysis.

**Figure 5 molecules-27-04390-f005:**
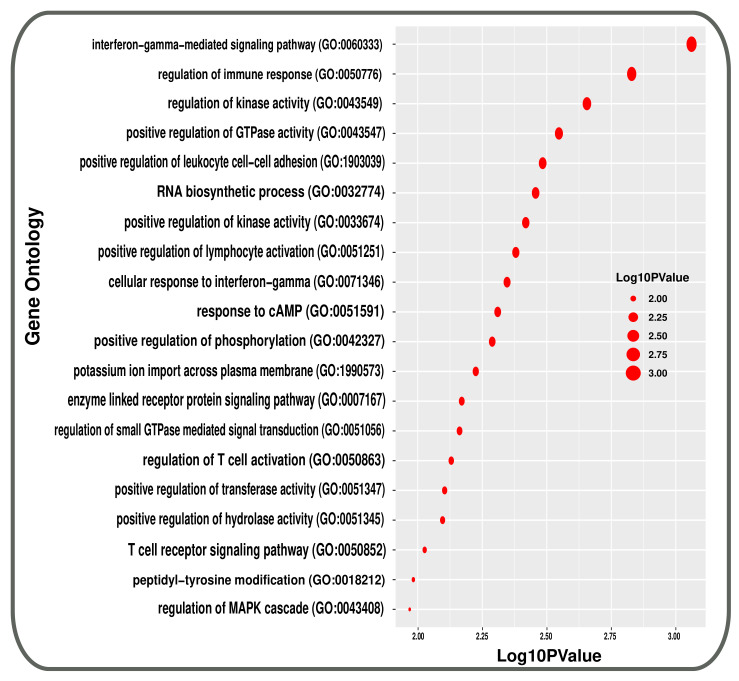
Top 20 Gene Ontologies represented using a bubble plot that are linked with both conditions in transcriptomic analysis.

**Figure 6 molecules-27-04390-f006:**
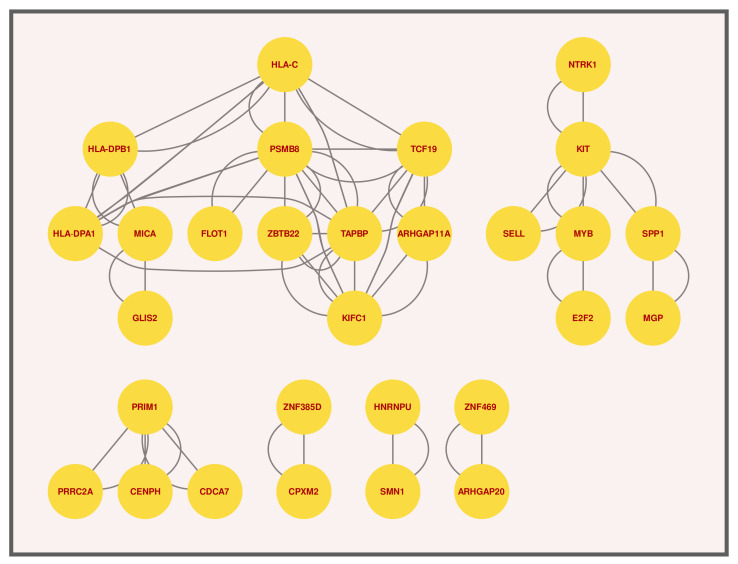
The figure illustrate the Protein–Protein Interactions between T2D and Smoking.

**Figure 7 molecules-27-04390-f007:**
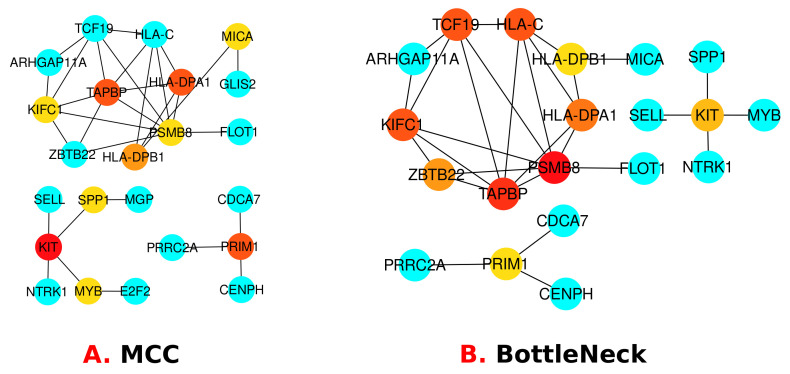
Hub proteins identified using 2 different cyto-hubba algorithms called MCC and BottleNeck to demonstrate the association between T2D and Smoking.

**Figure 8 molecules-27-04390-f008:**
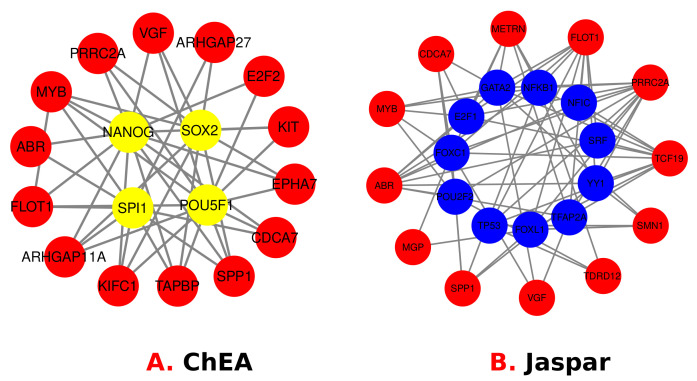
TF–Gene interactions showed using 2 different algorithms called ChEA and Jaspar to illustrate the linking between T2D and Smoking.

**Figure 9 molecules-27-04390-f009:**
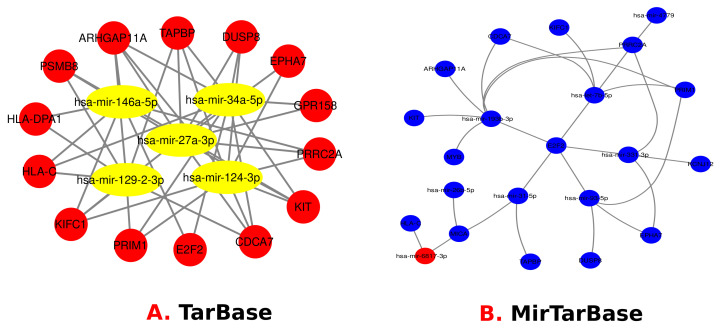
Gene miRNA is identified between T2D and Smoking using 2 different algorithm called TarBase and MirTarBase.

**Figure 10 molecules-27-04390-f010:**
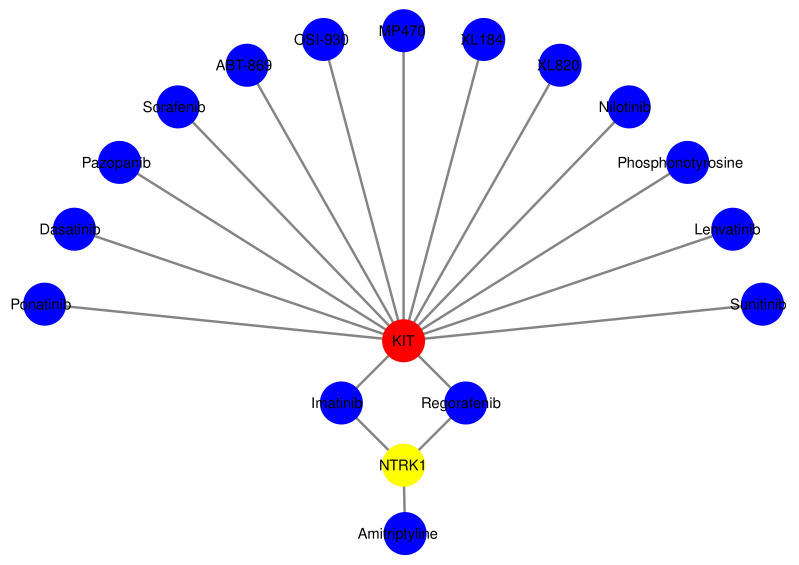
The figure shows the Protein–Drug Interaction between T2D and smoking using Protein–Drug Interaction algorithm.

**Figure 11 molecules-27-04390-f011:**
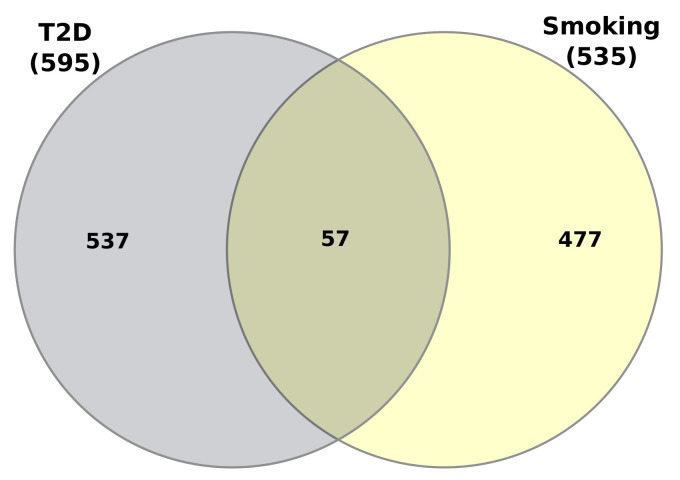
The Venn Diagram represented the shared candidate/biomarker genes found between Type 2 diabetes and smokers from GWAS studies.

**Figure 12 molecules-27-04390-f012:**
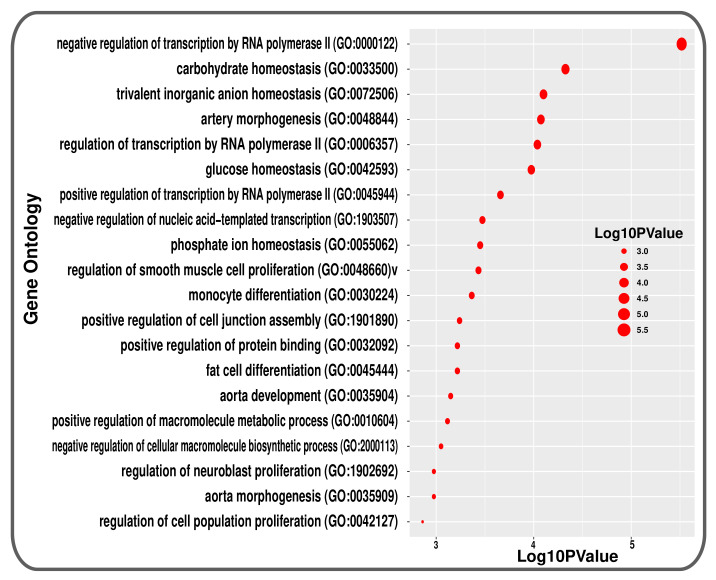
Top 20 GOs are represented using a bubble plot that are associated with both conditions in GWAS analysis.

**Figure 13 molecules-27-04390-f013:**
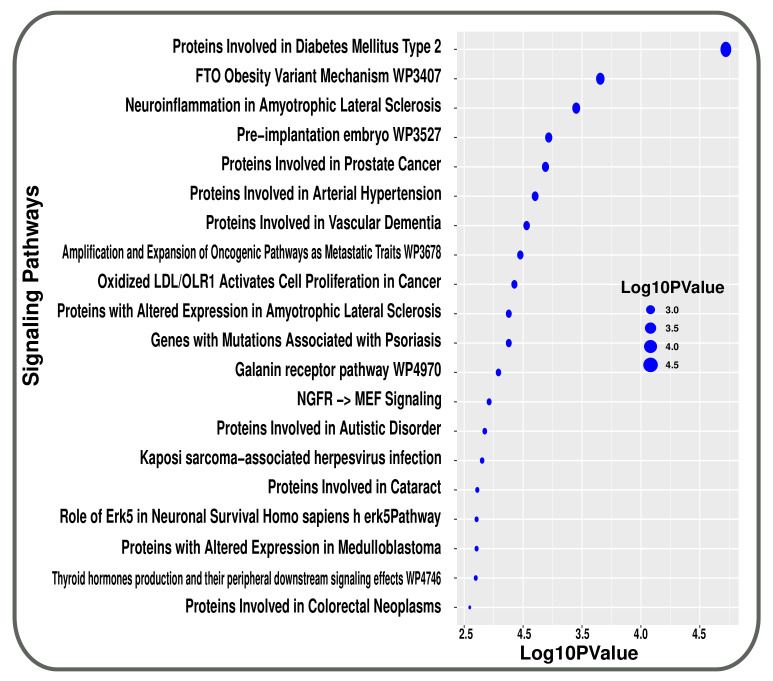
Top 20 pathways are represented using a bubble plot that are associated with both conditions in GWAS analysis.

**Table 1 molecules-27-04390-t001:** An overview of the data and findings from the transcriptome study. It includes the dataset accession number, sample source, total raw genes, sample number, significant genes.

Disease Name	GEO Platform	Tissues/Cells	GEO Accession	RAW Genes	Case Samples	Control Samples	Significant	Up Reg. Genes	Down Reg. Genes
Type-2 Diabetes (T2D)	Illumina NextSeq 500 (Homo sapiens)	Human cardiac mesenchymal cells	GSE-106177	18,619	7	7	1367	768	599
Smoking	Illumina HiSeq 2000 (Homo sapiens)	Airway Basal Cells	GSE-47718	21,178	7	10	962	682	280

**Table 2 molecules-27-04390-t002:** Identified possible target genes or biomarkers that are shared by smoking and type 2 diabetes have been confirmed by previous studies.

Gene	Smoking	T2D
SPP1	[56]	[57]
MICA	[58]	-
PSMB8	[59]	[60]
XIST	-	[61]
DUSP8	-	[62]
ABR	-	[63]
PRRC2A	[65]	[64]
NTRK1	-	[66]
EPHA7	[67]	-
SELL	-	[68]
PRIM1	[70]	[69]
KIFC1	[71]	-
ZNF385D	[72]	-
HLA-DPB1	-	[73]

## Data Availability

Links of our dataset are included in this manuscript.

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
