# Peer review of "Statistical Bioinformatics to Uncover the Underlying Biological Mechanisms That Linked Smoking with Type 2 Diabetes Patients Using Transcritpomic and GWAS Analysis"

_molecules, 2022, doi:10.3390/molecules27144390_

Round 1
Reviewer 1 Report
1. Abstract is too long. Please shorten the abstract as per Journal's requirements. Please focus in the abstract on the main scope of your study and what added value it adds to the current knowledge.
2. The introduction is too long, and its focus is not clear. It should be much shorter, having only three paragraphs, explaining the background, previous studies and clearly explaining the aim of the present study. I strongly suggest that a clinician revises this part, as the diabetes and smoking part are quite clumsily explained, mixing the terms, and suggesting some inconsistencies and wrong quotes.
As for the figure one, all the abbreviations must be explained in the figure caption.
3. Table captions should be above the table. All the abbreviations should be explained.
4. Discussion summarizes again all the findings but does not contemplate on their implication. Also, referring to smoking, as a disease, is strong. This should be elaborated if author want to continue using this term.
I would suggest rewriting the Discussion, explaining how these finding are of use in better understanding of bot smoking and diabetes. I miss the proper aim of the study. Also, clinical utilisation at least should be mentioned.
Author Response
Reviewer #1, Concern #1:
- Abstract is too long. Please shorten the abstract as per Journal's requirements. Please focus in the abstract on the main scope of your study and what added value it adds to the current knowledge.
>>Author response: Thank you for your suggestions. We have shortened our abstract part according to your suggestions.
Reviewer #1, Concern #2:
- The introduction is too long, and its focus is not clear. It should be much shorter, having only three paragraphs, explaining the background, previous studies and clearly explaining the aim of the present study. I strongly suggest that a clinician revises this part, as the diabetes and smoking part are quite clumsily explained, mixing the terms, and suggesting some inconsistencies and wrong quotes.
>>Author response: Thanks for your detailed comments. According to your comments, we have revised and modified our introduction, which is highlighted in the manuscript.
Reviewer #1, Concern #3:
- As for the figure one, all the abbreviations must be explained in the figure caption. Table captions should be above the table. All the abbreviations should be explained.
>>Author response: Thank you for your observations. All the abbreviations are explained following your comments.
Reviewer #1, Concern #4:
- Discussion summarizes again all the findings but does not contemplate on their implication. Also, referring to smoking, as a disease, is strong. This should be elaborated if author want to continue using this term. I would suggest rewriting the Discussion, explaining how these finding are of use in better understanding of both smoking and diabetes. I miss the proper aim of the study. Also, clinical utilisation at least should be mentioned.
>>Author response: Thank you for your comments. The discussion is rewritten according to your comments, which is highlighted in the manuscript.
Reviewer 2 Report
The authors describe a complete analysis of differentially expressed genes that are shared between two independent experiments gathered from GEO database (not performed by the authors). One experiment is about Type 2 Diabetes (T2D) and thew other is about smoking individuals. The general objective is to detect shared genes, pathways, Gene Ontology terms, Protein-Protein Interactions, Protein-Transcription Factors Interactions, Protein-Drug Interactions, Gene-miRNA Interactions and biomarkers (mutations).
The work is very descriptive: all analysis are carried-out using standard analytical tools (well chosen) and the results are presented as list of genes or bioentities that are shared between T2D and smoking. Some descriptive figures are provided with numerical scores of the bioentities detected in each type of analysis.
However, there is no discussion about the relevance of the findings or their novelty. It is difficult to make relationships between different types of results as lists are presented embebed in the text instead as tables. Discussion section is just a summary of the performed steps (and contains several errors when citing the figures) not a real discussion. Conclusion section is also of poor quality.
In summary, although the bioinformatic analysis are technically correct (tools and thresholds are standard), the authors are transformed data into information (lists of items), but no new results or useful conclusions are obtained from this information. Only a brief comment about genes mentioned in this paper that also appeared previously in literature is included.
Particular issues detected:
General:
English can be improved. Many details and basic descriptions of databases or methods are repeated several times in different sections. The content can be reduced by eliminating these redundancies.
Figures 1, 11, 12 and 13 are not mentioned in the text.
In Introduction
In the figure 1 there is a typo: "Validated bye previous literature"
In section 2.2
"Preprocessing of Raw genes" should be "Preprocessing of Raw counts"
Typo: "absulate" instead of "absolute"
In section 2.4
"GWAS Catalogue" should be "GWAS Catalog"
In section 3.2
"489 GO terminologies are found as the mostly enriched pathways between the conditions" should be "489 GO terminologies are found as the mostly enriched GO terms between the conditions"
In Discussion:
Mention to FIGURE-1 actually refers to FIGURE-2
Mention to FIGURE-2 actually refers to FIGURE-3
Mention to FIGURE-3 actually refers to FIGURE-4
Mention to FIGURE-4 actually refers to FIGURE-5
Author Response
The authors describe a complete analysis of differentially expressed genes that are shared between two independent experiments gathered from GEO database (not performed by the authors). One experiment is about Type 2 Diabetes (T2D) and thew other is about smoking individuals. The general objective is to detect shared genes, pathways, Gene Ontology terms, Protein-Protein Interactions, Protein-Transcription Factors Interactions, Protein-Drug Interactions, Gene-miRNA Interactions and biomarkers (mutations).
The work is very descriptive: all analysis are carried-out using standard analytical tools (well chosen) and the results are presented as list of genes or bioentities that are shared between T2D and smoking. Some descriptive figures are provided with numerical scores of the bioentities detected in each type of analysis.
Reviewer #2, Concern #1:
However, there is no discussion about the relevance of the findings or their novelty. It is difficult to make relationships between different types of results as lists are presented embebed in the text instead as tables. Discussion section is just a summary of the performed steps (and contains several errors when citing the figures) not a real discussion. Conclusion section is also of poor quality.
In summary, although the bioinformatic analysis are technically correct (tools and thresholds are standard), the authors are transformed data into information (lists of items), but no new results or useful conclusions are obtained from this information. Only a brief comment about genes mentioned in this paper that also appeared previously in literature is included.
>>Author response: Thank you for your comments. We have modified our introduction, discussion and conclusion section explaing the aim and relevance/novellity of our findings according to your comments which are highlighted.
Reviewer #2, Concern #2:
Particular issues detected:
General:
English can be improved. Many details and basic descriptions of databases or methods are repeated several times in different sections. The content can be reduced by eliminating these redundancies.
Figures 1, 11, 12 and 13 are not mentioned in the text.
In Introduction
In the figure 1 there is a typo: "Validated bye previous literature"
In section 2.2
"Preprocessing of Raw genes" should be "Preprocessing of Raw counts"
Typo: "absulate" instead of "absolute"
In section 2.4
"GWAS Catalogue" should be "GWAS Catalog"
In section 3.2
"489 GO terminologies are found as the mostly enriched pathways between the conditions" should be "489 GO terminologies are found as the mostly enriched GO terms between the conditions"
In Discussion:
Mention to FIGURE-1 actually refers to FIGURE-2
Mention to FIGURE-2 actually refers to FIGURE-3
Mention to FIGURE-3 actually refers to FIGURE-4
Mention to FIGURE-4 actually refers to FIGURE-5
>>Author response: Thank you for your comments. And we have revised our manuscript to make it more readable, replaced the line with appropriate words and improved English in the revised manuscripts, which are highlighted.
Round 2
Reviewer 1 Report
The article is now better, but the Discussion is too long and extensive. There is still a lack of clear messages. I strongly advise shortening the Discussion and focusing on the main points of the research, and not copying the descriptions from the Methods. Clinical utilisation is still too weak, the message should be strong. Avoiding smoking is a crucial factor obviously, so it should be described in that mind, as developing drugs for smokers and nonsmokers would be irrational... Also, English editing is necessary.
Author Response
Concern #1:
- The Discussion is too long and extensive. There is still a lack of clear messages. I strongly advise shortening the Discussion and focusing on the main points of the research, and not copying the descriptions from the Methods.
Author Response: Thank you for your suggestions. We have shortened our discussion with clear messages and focused on our research's main points, which are highlighted in the manuscript according to your suggestions.
Concern #2:
- Clinical utilisation is still too weak, the message should be strong. Avoiding smoking is a crucial factor obviously, so it should be described in that mind, as developing drugs for smokers and nonsmokers would be irrational... Also, English editing is necessary.
Author Response: Thanks for your comments. We have tried to strengthen the clinical utilisation with good rationale about developing drugs that are also highlighted in the manuscript.
We massively checked overall English quality and modified accordingly. Hope this version would be better from previous.
Reviewer 2 Report
Authors addresses the issues, specially the discussion section that was my main concern in the first review.
Author Response
We are grateful to the reviewer for this comments.
This manuscript is a resubmission of an earlier submission. The following is a list of the peer review reports and author responses from that submission.